The effects of soil drought stress on growth characteristics, root system, and tissue anatomy of Pinus sylvestris var. mongolica

Meng Fanjun
Zhang Tianze
Yin Dachuan yindachuan@syau.edu.cn
College of Forestry, Shenyang Agricultural University , Shenyang , China
Dąbrowski Piotr
Electronic publication date: 2023 Jan 9
Publication date: 2023
Volume: 11
Electronic Location ID: e14578
Received 2022 Sep 7; Accepted 2022 Nov 28
Copyright: ©2023 Meng et al.
Copyright year: 2023
Copyright holder: Meng et al.
License: This is an open access article distributed under the terms of the Creative Commons Attribution License, which permits unrestricted use, distribution, reproduction and adaptation in any medium and for any purpose provided that it is properly attributed. For attribution, the original author(s), title, publication source (PeerJ) and either DOI or URL of the article must be cited.
License URL: https://creativecommons.org/licenses/by/4.0/

Keywords: Physiological response, Drought tolerance, Morphological structure, Growing status

Funding: National Natural Science Foundation of China 31800542 This work was supported by the National Natural Science Foundation of China (31800542). The funders had no role in study design, data collection and analysis, decision to publish, or preparation of the manuscript.

==============================
The main purpose of this study was to study the changes in growth, root system, and tissue anatomical structure of Pinus sylvestris var. mongolica under soil drought conditions. In this study, the growth indexes and photosynthesis of P. sylvestris var. mongolica seedlings under soil drought stress were studied by pot cultivation. Continuous pot water control experiment of the indoor culture of P. sylvestris var. mongolica was carried out, ensuring that the soil water content of each treatment reached 80%, 40%, and 20% of the field moisture capacity as control, moderate drought and severe drought, respectively. The submicroscopic structures of the needles and roots were observed using a scanning electron microscope and a transmission electron microscope. The response of soil roots to drought stress was studied by root scanning. Moderate drought stress increased needle stomatal density, while under severe drought stress, stomatal density decreased. At the same time, the total number of root tips, total root length, root surface area, and root volume of seedlings decreased with the deepening of the drought. Furthermore, moderate drought and severe drought stress significantly reduced the chlorophyll a and chlorophyll b content in P. sylvestris var. mongolica seedlings compared to the control group. The needle cells were deformed and damaged, and chloroplasts and mitochondria were damaged, gradually disintegrated, and the number of osmiophiles increased. There was also an increase in nuclear vacuolation.

Introduction

Drought is a major environmental factor that restricts plant growth and development and even affects the distribution pattern of the world’s forests (Sherwood et al., 2013). There will be more severe and frequent droughts in many parts of the world as a result of climate change (IPCC, 2013; Trenberth et al., 2014; Okunlola et al., 2017). Drought is a climate disaster that occurs under most climatic conditions and can have considerable economic, social, and environmental impacts. In recent decades, drought caused by rapid warming has deeply affected the global forest ecosystem (Allen, Macalady & Chenchouni, 2010; Vurukonda et al., 2016; Nolan et al., 2018; Anderegg William et al., 2019).

Pinus sylvestris var. mongolica is an important geographical variety of P. sylvestris in eastern Asia. It has the biological characteristics of strong resistance to stress, rapid growth, and rapid lumber. Therefore, due to these properties, it has become one of the main afforestation tree species for vegetation restoration in mountainous, grassland and sandy land in arid and semi-arid areas of northern China. This plant species plays an important role in ecological construction and environmental restoration, such as soil and water conservation, wind prevention, and sand fixation (Zhu et al., 2003; Song et al., 2018; Guo et al., 2019; Kong et al., 2019). However, since the 1990s, the artificial sand fixing pure forest of P. sylvestris var. mongolica has shown a recession phenomenon. The decline was manifested mainly in withered and yellow branches, reduced growth, the occurrence of diseases and pests, leading to the death of the whole plant and could not be natural regeneration (Meng et al., 2010; Song, Zhu & Zheng, 2017). Thus, considering the recession mechanism of P. sylvestris var. mongolica, the current study focuses mainly on water factors (Song, Zhu & Kang, 2013), and it was found that soil water and drought stress are the main factors limiting its development and growth in sandy soil (Song, Zhu & Yan, 2015).

Drought stress has many effects on plant growth and metabolism. Initially, drought stress directly affects the germination of plant seeds and reduces the survival rate of seedlings (Shi, Ding & Yuan, 2004; Wei, 2005). Secondly, drought stress directly leads to a water deficit in plant cells, making them unable to divide and normally increase, thus inhibiting plant growth and development (Long & Deng, 2019). The study found that severe drought caused obvious drought damage symptoms to forest trees in northern Finland (Muukkonen et al., 2015). Drought also restricted the growth and development of Scots pine trees (Zang, Pretzsch & Rothe, 2012) and seriously reduced the radial growth of Picea crassifolia, Pinus tabuliformis, Larix decidua Mill. and Picea meyeri (Vitasse et al., 2019; Gao, Yang & Qin, 2021; Xue et al., 2022a; Xue et al., 2022b). At the same time, previous studies have shown that similar larch species (such as Larix principis-rupprechtii Mayr) are at high risk of growth stagnation in large central and northern China areas during extreme drought (Zhang et al., 2021).

The root structure determines the efficiency of water absorption and transport of plants, which can help alleviate the damage caused by drought stress. Under drought stress, the root tip first perceives the signal and transmits it to the aboveground part (Jia & Zhang, 2008; Yin, Wang & Qi, 2021). After drought, phenotypic characteristics such as total surface area, total volume, average diameter, total length, and biomass of plant roots change (Zhang & Sun, 2016; Zhang et al., 2019). In addition, the morphology and number of mitochondria and other organelles in root cells will also change (Liu, Yue & Chen, 2010). Under mild drought, plants can improve their tolerance to drought by increasing the length of the main roots and the number of lateral roots and root hairs (Salazar-Henao, Vélez-Bermúdez & Schmidt, 2016). However, with the deepening of drought, plant root respiration is reduced, resulting in an insufficient supply of ATP (adenosine triphosphate) and a significant decrease in root activity that leads to slowing or even stops the growth (Kim, Chae & Choi, 2020; Nikolova et al., 2020). The water balance in plants also alters, causing irreversible damage to the plants (Ma et al., 2012; Isaji et al., 2018). Some studies found that under short-term drought stress, Robinia pseudoacacia absorbs more water by promoting the relative growth of fine roots. However, under long-term drought stress, the root growth of Robinia pseudoacacia is inhibited (Gao, Wang & Zhang, 2010). At the same time, moderate drought treatment will increase the root biomass of Platycladus orientalis and Pinus tabulaeformis while decreasing under severe drought stress (Chen & Zhao, 2011). And different degrees of water loss also affect the root water content and root activity of Larix principis-rupprechtii and Pinus tabuliformis (Chen, Gao & Shi, 2017). Similarly, changes in root microstructure can also reflect the adaptability of plants to drought stress. The structure and properties of the root cortical tissue, the diameter, and the number of xylem vessels changed according to the degree of drought stress (Konijnendijk & Randrup, 2002). Furthermore, some studies have also found that drought stress reduced the average root diameter and the diameter of the root vessel (Wang, Zhang & Liu, 2005; Lee et al., 2016; Wang et al., 2018). Previous studies showed that long-term drought stress reduced the root length and root volume of Pinus sylvestris var. mongolica, and summer drought would limit the establishment of the Scot pine (Pinus sylvestris L.) forest by reducing growth and increasing seedling mortality (Castro et al., 2005; Qian et al., 2021).

The leaf structure is the most intuitive embodiment of the adaptability of plants to arid habitats. It determines the functions of plant carbon assimilation, water loss, and retention and can also be used to evaluate biomass accumulation capacity (Marron, Dillen & Ceulemans, 2007). To adapt to drought, plant needles tend to increase mesophyll palisade tissue and the number of cell layers and reduce spongy tissue, cell volume, and cell space (Chartzoulakis et al., 2002; Burling et al., 2013; Scoffoni et al., 2014; Bhusal et al., 2020). In addition, the self-regulation ability of the stomata in the needles can also reflect the drought resistance of plants to some extent (Guo & Wu, 2015). Plants resistant to drought can regulate the stomata to a greater degree to reduce water loss (Fiorin, 2016). Under drought stress, the relative water content of plants decreases, and the stomatal aperture decreases or even closes to reduce the water loss from needles, which facilitates the recovery of leaf water potential (Casson & Hetherington, 2009; Wang et al., 2010). Similarly, the reduction of stomatal conductance or stomatal closure of plant needles in an arid environment affects the absorption of CO2 and reduces the photosynthetic rate (Reddy, Chaitanya & Vivekanandan, 2004; Pagter, Bragato & Brix, 2005). In the context of chlorophyll, the main pigment responsible for photosynthesis, drought stress can cause chlorophyll decomposition and decrease chlorophyll content. It has been shown that drought stress causes chlorophyll decomposition and chlorophyll content decreases, leading to changes in photosynthetic function (Zhao et al., 2006; Jafari, Hashemi Garmdareh & Azadegan, 2019).

To reveal the decline mechanism of P. sylvestris var. mongolica, explaining the internal mechanism and the water influence mechanism on the plantation are key factors and provide a theoretical basis for the management of P. sylvestris var. mongolica plantation (Song, Zhu & Zheng, 2017). In this study, the potted water control experiments of P. sylvestris var. mongolica seedlings were conducted, and the growth characteristics of P. sylvestris var. mongolica seedlings were clarified by measuring changes in growth indexes, photosynthetic pigment indexes, root indexes, and submicroscopic structure under different levels of drought stress. The physiological mechanism of drought resistance in P. sylvestris var. mongolica seedlings was briefly explained, which provided a theoretical basis and practical reference for further study of the physiological mechanism under the drought-induced decline of P. sylvestris var. mongolica plantation.

Material and Methods

Plant materials and experimental design

The seeds of P. sylvestris var. mongolica were collected from Zhanggutai Experimental Forest Farm, Fuxin City, Liaoning Province. The seeds used in the experiment were disinfected and sterilized with potassium permanganate (0.5%, v/v) for 30 min, followed by washing with distilled water three times. In addition, seeds were wrapped with sterile gauze for moisturizing and kept under 25 °C for germination, sprayed with sterile water every day until growth. Seeds were transferred to plastic pots filled with sterilized vermiculite: soil: sand mixture (1:2:1) and kept under the corresponding controlled greenhouse conditions (Yin, Wang & Qi, 2021). After the seedlings were unearthed, fix the seedlings at 8 in each pot. Furthermore, three months after emergence, 30 pots of seedlings with stable and similar growth were selected and divided into three groups containing 10 pots in each group. In this experiment, three different treatments of drought were adopted; control (80% of field water capacity; CK), moderate drought (40% of field water capacity; MD), and heavy drought (20% of field water capacity; HD) (Yan et al., 2003; Zhu, Kang & Li, 2006; Shan et al., 2007). After emergence, each group of P. sylvestris var. mongolica seedlings was subjected to the corresponding drought stress treatment. The soil water content was maintained by weighing and replenishing water, i.e., each pot was weighed every day to supplement the lost water to maintain the stability of the corresponding soil water content.

Growth of seedlings

In the process of seedling drought treatment, the height and ground diameter of five random P. sylvestris var. mongolica seedlings per treatment were measured with a ruler and a Vernier caliper and recorded every week. After two months of drought treatment, the seedlings were dug out and cleaned with water. The fresh weight of the primary roots, lateral roots, stems, and needles of the seedlings was weighed, followed by drying in the oven to a constant weight. The dry weight of each part was measured using balance.

Stomatal density of needles

The needles of P. sylvestris var. mongolica treated with different treatments were sliced, and their slides were observed and photographed under the electron scanning microscope (Hitachi s-3400n). The number of pores was observed in five visual fields, and the number of pores in each visual field was counted using Image J software, and the density of the pores was calculated.

Chlorophyll content in seedlings

The chlorophyll content of the seedlings was measured as described by William & Paul (1985). Fresh leaf samples were cleaned with deionized water. After the surface pollution was removed, 0.5 g of needles were added to 10 ml of acetone for grinding. In addition, the samples were centrifuged at 10,000 rpm for 5 min and the supernatant was collected, followed by spectrophotometer analysis at 663 nm and 645 nm. The experiment was repeated three times.

Root morphological indexes of seedlings

The entire root system was carefully separated from the soil, cleaned with tap water and deionized water, and then scanned with a root scanner (Epson Expression 1640XL scanner, Epson, Quebec, Canada). The length, surface area, volume, and the number of roots at the tip were analyzed by WinRhizo Reg software.

Microscopic observation of seedlings

Fresh samples of roots, stems, and needles from P. sylvestris var. mongolica seedlings were rinsed and cut into small samples of one cm, then immediately fixed in 4% glutaraldehyde solution. The samples were put under vacuum suction until completely immersed in the fixative and kept at 4 °C overnight. Furthermore, the samples were rinsed with phosphate buffer (0.1M, pH 6.8) 3–5 times, 10-15 min each time. The samples were then fixed in 1% osmic acid for 2 h and then transferred to phosphate buffer (0.1 M, pH 6.8) for 1 h. A gradient of acetone solution (30%, 50%, 70%, 80%, 90%, and 100% v/v) was used to dehydrate twice for 15–30 min each time. The samples were then treated twice with isoamyl acetate for 30 min and 20 min, respectively, and shaken to replace acetone in the sample. Furthermore, the samples were collected in a sample cage and placed in a critical point dryer (HCP-2, Hitachi, Tokyo, Japan) for drying. Finally, the stick and spray gold were performed with the JSM-6360LV scanning electron microscope (SEM) observation.

In each treatment, three groups of fresh needle samples of P. sylvestris var. mongolica seedlings were taken and rinsed. Then they were prefixed with 2.5% glutaraldehyde overnight at 4 °C. After being washed three times with phosphate buffer saline for 15 min each time, 1% osmic acid fixative was used for fixation and was kept for 2 h. Furthermore, after being fully washed with PBS and ethanol (30%, 50%, 70%, 90%, and 100%), samples were successively used for dehydration. Then the ethanol was replaced with 25%, 50%, 75%, and 100% propylene oxide (dissolved in ethanol). A gradient of resin embedding agent (10%, 30%, 70%, and 100%, dissolved in propylene oxide) was used for penetration and, finally, 100% resin embedding agent was used to polymerize at 70 °C for 12 h (Wei, Wang & Zhang, 2010). A Leica EM UC slicer was used to slice ultrathin sections, and uranyl acetate-lead citrate double staining was performed. Finally, the H-7650 transmission electron microscope (TEM) was used to observe and photograph the samples. A total of nine slices were observed from three samples in each group.

Statistical analysis

SPSS software was used for the one-way analysis of variance (ANOVA). To illustrate the graph, Excel 2010 and GraphPad Prism 7.0 software was used for chart making.

Results

Analysis of plant growth parameters

The growth of P. sylvestris var. mongolica seedlings under drought treatment was inhibited both during and after the experiment (Fig. 1). It is clear from Fig. 2 that under different degrees of drought stress, the difference in seedling height and ground diameter between CK and MD as well as HD gradually increased with time. During the fifth week of drought treatment, the seedling height and ground diameter in the CK increased by 42.96% and 26.7%, respectively, compared to those in the initial state. Similarly, the seedling height and ground diameter ratio of MD treatment increased by 27.18% and 15.35%, respectively; however, in the HD treatment, these parameters increased only by 21.48% and 9.19%, respectively. At the end of the drought treatment, which was the eighth week, the seedling height and ground diameter in the CK increased by 65.93% and 37.36%, respectively, compared to the initial value. These parameters increased by 43.9% and 22.68% in MD treatment, respectively. However, in HD treatment, they increased only by 30.28% and 12.56%, respectively (Figs. 2A and 2B). These findings indicate that drought inhibited the growth of plant height and ground diameter and the degree of inhibition of seedling height and ground diameter increased with the increase in water deficit and time.

Figure 1 Growth of P. sylvestris var. mongolica under different drought conditions.

CK, non-drought stress; MD, moderate drought; HD, heavy drought.

Figure 2 Changes in the seedling height and ground diameter of P. sylvestris var. mongolica under different drought stress; ((A) Seedlings height; (B) Ground diameter; CK, non-drought stress; MD, moderate drought; HD, heavy drought).

The lowercase letters indicate significant (p < 0.05) differences between different treatments at the same time.

Biomass analysis of different tissue

The effects of various degrees of drought stress on the biomass of P. sylvestris var. mongolica seedlings are shown in Figs. 3 and 4. For MD and HD-treated seedlings, the fresh weight of each part was significantly different from CK, which showed that the fresh weight of the primary root, the lateral root, the stem, and the leaf decreased significantly after drought treatment (P < 0.05). Compared to CK, the primary root and lateral root of P. sylvestris var. mongolica seedlings decreased by 39.29% and 33.75% under MD treatment, 60.71%, and 57.50% under HD treatment, respectively (Fig. 3A). The decrease in the stem was 25% and 40.91% under MD treatment, and HD treatment, respectively (Fig. 3B), and leaf decreased by 13.57% and 51.43% under MD treatment and HD treatment, respectively (Fig. 3C). In case of the dry weight of each tissue after drought treatment, the lateral roots of the plant both under MD treatment and HD treatment were significantly lower than CK, while, for primary roots, stems, and needles, only under HD treatment had significant differences compared to CK (P < 0.05). The dry weights of the roots (primary roots and lateral roots), stems, and needles of the seedlings were significantly decreased under HD treatment. There was a decrease of 18.18% and 13.64% under MD treatment and 36.36% and 27.27% under HD treatment, respectively, in the dry weight of the primary root and the lateral root (Fig. 4A). Furthermore, the dry weight of the stem decreased by 14.29% and 21.43% compared to CK under MD treatment and HD treatment (Fig. 4B), and the leaf decreased by 8.33% and 31.25% compared to CK under MD treatment and HD treatment, respectively (Fig. 4C). This indicated that MD and HD treatment reduced the dry and fresh weight of plants and severely inhibited the growth and accumulation of dry matter in seedlings.

Figure 3 Fresh weight of each tissue of P. sylvestris var. mongolica seedlings after drought stress treatment.

(A) Root; (B) Stem; (C) Leaves; CK, non-drought stress; MD, moderate drought; HD, heavy drought. The lowercase letters indicate significant (p < 0.05) differences among different treatment times subjected to the same species. Error bars are ±SD (n = 6).

Figure 4 Dry weight of each tissue of P. sylvestris var. mongolica seedlings after drought stress treatment.

(A) Root; (B) Stem; (C) Leaves; CK, non-drought stress; MD, moderate drought; HD, heavy drought. The lowercase letters indicate significant (p < 0.05) differences among different treatment times subjected to the same species. Error bars are ± SD (n = 6).

Analysis of photosynthetic pigment content

Under various degrees of drought stress, the contents of chlorophyll a and chlorophyll b in the needles of P. sylvestris var. The mongolica seedlings showed a downward trend (Fig. 5). The chlorophyll a and chlorophyll b content in MD or HD treatment was significantly different from CK (P < 0.05). Under MD and HD treatment, chlorophyll a decreased by 18.31% and 61.11%, respectively, compared to CK (Fig. 5A), while chlorophyll b in seedlings decreased by 17.70% and 55.37%, respectively, in MD and HD (Fig. 5B). Furthermore, the chlorophyll a/b value under MD treatment decreased compared to CK, but there was no significant difference (P > 0.05). There was a significant difference between HD and CK (P < 0.05) (Fig. 5C).

Figure 5 Chlorophyll content of P. sylvestris var. mongolica seedlings after different drought stress treatments.

(A) Chla; (B) Chlb; (C) Chla/b; CK, non-drought stress; MD, moderate drought; HD, heavy drought. The lowercase letters indicate significant (p < 0.05) differences among different treatment times subjected to the same species. Error bars are ±SD (n = 6).

Analysis of stomatal density

The stomatal density of P. sylvestris var. mongolica needles under different drought stresses is shown in Fig. 6. Compared to CK, MD treatment showed an increasing trend. The stomatal density increased by 4.09% under MD treatment and decreased by 3.02% under HD treatment compared to CK. However, these alterations were not prominent, and there were no significant differences in MD and HD treatment compared to CK by analysis of variance (P > 0.05).

Figure 6 Change of stomatal density in leaves of P. sylvestris var. mongolica under different drought stress.

CK, non-drought stress; MD, moderate drought; HD, heavy drought. The lowercase letters indicate the significant (p < 0.05) differences among different treatment times subjected to the same species. Error bars are ±SD (n = 5).

Analysis of root structure of seedlings

The root scanning diagram of P. sylvestris var. mongolica seedlings under various degrees of drought are shown in Fig. 7. The measurements of root growth parameters under different drought treatments are shown in Table 1. It is clear from Fig. 7 and Table 1 that HD treatment significantly reduced total root length, surface area, root volume, average root diameter, total root length per unit of soil volume (LenPerVol), root tips number, root fork number, and crossing number of P. sylvestris var. mongolica (P < 0.05). Similarly, MD treatment significantly reduced root surface area, root volume, total root length per unit of soil volume, root fork number, and crossing number of P. sylvestris var. mongolica (P < 0.05). Compared to CK, the total root length of MD and HD treatment decreased by 9.79% and 27.88%, respectively; the root surface area decreased by 6.62% and 8.89%, respectively; the root volume decreased by 13.07% and 38.34%, respectively; the total root length per unit of soil volume decreased by 14.10% and 18.42%, respectively; the number of root tips decreased by 8.40% and 18.42%, respectively; the root forks number decreased by 15.07% and 22.87%, respectively, and the number of crossings decreased by 18.28% and 29.66%, respectively.

Figure 7 Effects of drought stress on the root structure of P. sylvestris var. mongolica seedlings.

Analysis of the microstructure under the scanning electron microscope

The microscopic study of plant needles can reveal the effect of drought stress more intuitively on the morphology of P. sylvestris var. mongolica needles. Adaptive changes in leaf structural characteristics are an important manifestation of the plant response to drought. With increasing drought stress, starch grains in the needles of P. sylvestris var. mongolica gradually decreased (Figs. 8A–8C); the elevated drought stress forced the stomatal opening to decrease (Figs. 8G–8I). Furthermore, with increasing drought stress, the epidermis of the needles appeared to fold and with distortion (Figs. 8D–8F). In the CK, the cross-sectional shape of the needle showed a relatively full state as a whole, and the structure of each part was clear. MD treatment caused a slight shrinkage of transmission tissue cells, and the needles were deformed. On the other hand, under HD treatment, the needles were severely shrunk and deformed, the shrinkage of the transmission tissue cells intensified, and the area decreased significantly (Figs. 8P–8R). Simultaneously, with increasing drought stress, the tracheid diameter of the root of P. sylvestris var. mongolica shrinks and gets injured (Figs. 8J–8L). Similarly, the tracheid diameter of the stem becomes smaller and damaged (Figs. 8M–8O).

Table 1 Root system index of P. sylvestris var. mongolica seedlings after different drought stress treatments. (LenPerVol: total root length per unit soil volume).

Drought intensity Treatments (T)	Length (cm)	Surface area (cm2)	Root volume (cm3)	Average diameter (mm)	LenPerVol (cm/m3)	Tips	Forks	Crossings	
CK	898.19 ± 11.72a	575.41 ± 9.18a	34.06 ± 0.88a	2.88 ± 0.3a	888.85 ± 17.17a	4,332 ± 105a	9,324 ± 250a	1,116 ± 64a	
MD	810.17 ± 30.26a	537.33 ± 7.81b	29.61 ± 0.85b	2.53 ± 0.12a	763.50 ± 27.32b	3,968 ± 126a	7,919 ± 186b	912 ± 36b	
HD	647.75 ± 35.62b	524.27 ± 2.03b	21 ± 0.32c	1.90 ± 0.18b	677.42 ± 16.08b	3,534 ± 160b	7,192 ± 121b	785 ± 14b	
Notes.

The lowercase letters indicate the significant (p < 0.05) differences among different treatment times subjected to the same species. Error bars are ±SD (n = 3).

Figure 8 SEM observation of P. sylvestris var. mongolica under different drought stress (CK, MD, and HD in each column from left to right).

(A–C) starch granule morphology in needles; (D–F) needle epidermis morphology; (G–I) stomatal morphology of needles; (J–L) transverse anatomical structure of root; (M–O) transverse anatomical structure of stem; (P–R) anatomical structure of needle transverse section).

Microscopic tissue analysis under transmission electron microscopy

To further observe the ultrastructure of P. sylvestris var. mongolica needle cells, the submicroscopic structure of the needle cells was analyzed using a transmission electron microscope (Fig. 9). It is clear from Fig. 9 that under CK, the mesophyll cells of P. sylvestris var. mongolica were filled and closely arranged; the morphology of mesophyll cells was normal (Fig. 9G); the chloroplast structure in the cells was spindle-shaped; the chloroplasts of the cells were attached to the inner wall of the mesophyll cells in a spindle shape and were arranged neatly; and starch granules, as well as a small amount of osmiophilic globule on the chloroplasts, can be observed (Fig. 9A). The distribution of the chloroplast stroma was compact, the chloroplast bilayer membrane was visible, and the thylakoid lamellar structure was developed, clearly visible, and arranged in parallel (Fig. 9D). The structure of mitochondria in cells was complete and abundant, and the matrix was evenly distributed in the mitochondria. The inner and outer membranes of the mitochondria were intact, and the cristae of the mitochondria could also obviously be observed (Fig. 9M). The nuclear structure was complete, and the nucleoplasm compact (Fig. 9J). Under MD treatment, the structure of the chloroplast bilayer membrane was destroyed, and the thylakoid lamella was reduced and blurred. Starch granules were still observed in the chloroplasts, and the number of osmiophilic globules increased (Figs. 9B, 9E and 9H). The outer membrane of the mitochondria showed that the cristae of the mitochondria began to expand and that the whole mitochondria swelled and decreased in number (Fig. 9N). The nucleoplasm of the nucleus was loose (Fig. 9K). Furthermore, under HD treatment, the chloroplast bilayer membrane in the needle cells of P. sylvestris var. mongolica was destroyed, the starch granules wrinkled, and the outer wall blurred. The osmiophilic globule can no longer be identified, the chloroplast stroma flows out, the thylakoid disintegrates, and the lamellar structure of the thylakoid cannot be observed at all (Figs. 9C, 9F and 9I). The mitochondrial bilayer membrane was severely damaged, vacuolized, the mitochondrial matrix flowed out, and the cristae in the mitochondria had disappeared (Fig. 9O). The nuclear cytoplasm of the nucleus was severely vacuolated (Fig. 9L).

Figure 9 Ultrastructure of P. sylvestris var. mongolica needles under different drought stress (CK, MD, and HD in each column from left to right).

Chl, chloroplast; CM, chloroplast membrane; Th, thylakoid; CW, cell wall; N, nucleus; SG, starch.

Discussion

Water is a key limiting factor for seed germination and seedling growth and plant survival (Han, Cheng & Li, 2016). Drought stress is one of the common stresses in plant growth, and its impact on plants has been widely concerned (Yang, Miao & Xu, 2007).

Drought stress has many adverse effects on plants. Severe drought will even lead to the death of plants (Zhang et al., 2018). The effect of drought on the plant growth index is the most intuitive expression affected by drought stress. In this study, among the seedlings of CK, P. sylvestris var. mongolica seedlings have a higher degree of increase in plant height and ground diameter, indicating that under relatively good water conditions, plants maintained a high photosynthetic rate and thus ensured a relatively high carbon acquisition capacity (Fig. 10). However, under moderate or severe drought conditions, the photosynthetic rate, transpiration rate, and water use efficiency of plants decreased, which reduced the carbon acquisition ability of plants and inhibited the growth of plant height and ground diameter (Zhu et al., 2005). Overall, the growth of plant height and ground diameter in P. sylvestris var. mongolica seedlings gradually slow down with increasing drought stress, indicating that drought stress limits seedling growth, and these findings are consistent with previous research studies in Larix principis-rupprechtii, Pinus tabuliformis, Picea meyeri, and some other plant species (He, 2001; Xu, Guo & Xu, 2010; Xue et al., 2022a; Wang et al., 2022; Xue et al., 2022b). This also demonstrates that when plants are faced with drought stress, they normally limit their growth rate to reduce water consumption and maintain plant survival (Wu & Li, 2014; Poorter, Niklas & Reich, 2012). The dry weight and the fresh weight of the plant tissues declined as the drought deepened, especially under the condition of severe drought, and this effect was more pronounced. These results were consistent with the response of Larix gmelinii and many other plant seedlings to drought stress (Wang et al., 2020; Nadia et al., 2021; Park et al., 2021). In terms of biomass measured after drought stress, the root system was the first to be subjected to drought stress due to the different environments where the aboveground part and underground parts of the plant are located, and their ecological indicators, as well as their physiological and biochemical functions, directly affect the drought resistance of the plant. In an arid environment, the growth and development of plants, as well as their adaptability to the external environment, can be reflected by the structural characteristics of the roots (Pan, Zhang & Shao, 2017). The characteristic indexes of root growth of this study suggest that drought stress inhibited the indexes such as total root length, root surface area, root volume, total root length per unit of soil volume, bifurcation number, and crossing number (Fig. 10). Especially under severe drought, these indexes were significantly different from those of the control group. A reduction in total root length was observed, which was consistent with the results of tall fescue (Wang, Bughrara & Nelson, 2008). However, in moderate drought, there was no significant difference between total root length and root tip number with the control group, indicating that P. sylvestris var. mongolica can resist certain drought stress, but beyond that, changes appear in its morphological structure due to drought stress, thus affecting some physiological and biochemical functions (Fig. 10).

Figure 10 Effects of drought stress on growth of P. sylvestris var. mongolica.

Photosynthesis is essential for the growth and development of green plants, and chlorophyll is a primary substance involved in photosynthesis and the main pigment that drives photosynthesis. Therefore, to some extent, the rate of plant photosynthesis can be reflected by the chlorophyll content (Huang et al., 2012; Li et al., 2016; Zheng et al., 2018; Zhang et al., 2018). Chlorophylls a and b are the most common pigments involved in photosynthesis. In this process, chlorophyll a is mainly responsible for the conversion of light energy to chemical energy. In addition to absorption and conversion, chlorophyll b also has the responsibility of adjusting the size of the antenna for the photosynthetic mechanism (LaRoche et al., 1996; Green & Dumford, 1996; Barber, Morris & Büchel, 2000). When plants are subjected to drought or low-temperature stress, chlorophyll a decomposes faster than chlorophyll b, and the content ratio of both pigments changes, which can also reflect the rate of photosynthesis (Yamasato et al., 2005). Therefore, as the main participant in photosynthesis, the determination of its content is an important means to understand the rate of photosynthesis under drought stress (Fig. 10). The results of this study showed that the chlorophyll a and chlorophyll b content of P. sylvestris var. mongolica needles decreased significantly under drought stress. However, under severe drought stress, compared to moderate stress, the contents of chlorophyll a and chlorophyll b in the needles of P. sylvestris var. mongolica decreased greatly, which may be the compensation effect under moderate drought, or it may be because the plants were in the growing season and contain part of the undivided chlorophyll, so the reduction in chlorophyll content was relatively small. This conclusion is consistent with previous experimental studies on Mongolian pine, Scots pine, and some other plants (Zlobin et al., 2019; Xia et al., 2019; Zhang et al., 2020; Qian et al., 2021). Additionally, chlorophyll a and chlorophyll b of P. sylvestris var. mongolica seedlings gradually decreased with the aggravation of drought stress. In severe drought, the chlorophyll content was significantly different from that of the control group. Previous studies have shown that there is a negative correlation between chlorophyll a/b change and drought resistance (Zhao et al., 2019). This indicates the drought resistance of P. sylvestris var. mongolica plants gradually weakened under drought stress. The plant needles remain with the most contact with the outside world. In addition to serving as a crucial location for photosynthetic physiological processes, needles are not only an important part to ensure the normal growth and development of plants but are also a sensitive part of drought stress. Their morphological and structural changes are closely related to plant drought resistance (Li et al., 2013; Ren et al., 2015; Nina & Meruert, 2017). The size, opening, and density of the stomata on the needles can affect the rate of plant transpiration. Among them, stomatal density significantly affects plant drought tolerance, water use efficiency, and stomatal conductance. Increased stomatal density is also a typical feature of the plant response to drought stress (Fig. 10). A small and dense stoma will reduce transpiration and improve plant resistance to drought stress (Ferris et al., 1996; Drake, Froend & Franks, 2013; Guo & Wu, 2018). Furthermore, under mild drought, plant leaf cells elongate slowly, and leaf growth slows, decreasing leaf area and increasing stomatal density. In case of severe drought, the stomatal development of plant needles becomes slow, manifested in the decrease in stomatal density (Sam et al., 2000; Xu, Zou & Zhao, 2003; Xu & Zhou, 2008; Fraser et al., 2009). The findings of this study were also consistent with previous studies, as described. Furthermore, in moderate drought, the stomatal density increases, which may be due to the decrease in individual and leaf area and the increase in the number of stomata per unit area, resulting in the increase in the stomatal density (Xie, Song & Cao, 2015). This is consistent with previous research results by Larix kaempferi (Bhusal et al., 2020). The larger stomatal density is conducive to making full use of available water for photosynthesis and better control of respiration in a short time, contributes to heat dissipation, and reduces the degeneration of chloroplasts and protoplasts caused by drought. Under severe drought stress, the stomatal density decreases, which may be due to the needles being seriously affected, which inhibits the occurrence of stomata, significantly reduces the number of stomata, and finally shows the decrease of stomatal density (Wang, Ren & Lou, 1992).

The study of leaf microstructure is helpful in understanding the response of plants to drought stress and provides the basis for further research. The results of the scanning electron microscope showed that the size of stomata and opening of the needles were significantly reduced, which was consistent with the results of some previous studies on Cryptomeria japonica, Larix kaempferi and some other species of forest trees (Bhusal et al., 2020; Kenzo et al., 2021; Nadia et al., 2021). Furthermore, this study found that the tracheid diameter and tracheid wall thickness of roots, stems, and needles gradually decreased with the deepening of the drought. Previous studies have demonstrated that the tracheid diameter is positively correlated with water conveyance efficiency (Sperry, Hacke & Pittermann, 2006; Schuldt et al., 2016). Furthermore, the thickness of the root tracheids was closely related to embolism resistance. The thickening of the wall of the root tracheids can strengthen the fragile pipes of the roots and improve the mechanical strength of the roots, to improve the embolism resistance of roots (Fichot et al., 2010). This phenomenon shows that when subjected to drought stress, the water-carrying capacity and anti-embolism capacity of plants are inhibited. Meanwhile, in this study, with increasing degree of stress, phloem cells become denser and narrower, which is not conducive to the transportation and distribution of water and organic matter in plants, leading to plant growth restriction or even death due to the decline or unreasonable distribution of water, sugar and other organic matter. Transport tissue is primarily responsible for material transport between mesophyll cells and vascular bundles (Qi, 2006). Therefore, dehydration and deformation of cells in seedlings will also lead to the obstruction of material exchange between the two parts of tissues, which is not conducive to stable growth and metabolic activities of plants and could be one of the reasons why drought restricts the growth of P. sylvestris var. mongolica. Through the study of Picea mariana and Abies balsamea, it was found that the tracheid lumen size of trees was smaller and the cell wall was thicker under drought conditions, which was consistent with the results of this study (Belien et al., 2012; D’Orangeville et al., 2013).

Chloroplasts and mitochondria, important organelles performing more physiological functions, are sensitive to environmental changes. Their morphological structure and physiological functions often change under stress conditions (Vain, Pardha & Saradhi, 2001). The change in plant ultrastructure under drought stress is also an important index to measure its resistance to drought (Zhang, Li & Chen, 2014). According to the TEM analysis of this study, the number of chloroplasts and mitochondria in P. sylvestris var. mongolica needle leaf cells not only decreased with the aggravation of drought stress, but their morphological structure also changed to varying degrees. The bilayer membrane of chloroplasts and mitochondria was destroyed, the content flowed out, the lumen was hollowed out, the number of osmiophilic globules in chloroplasts increased under moderate drought, and the cristae in mitochondria also changed. Several studies have shown that drought stress can lead to the disordered arrangement and structural damage to chloroplasts, expansion, and rounding of mitochondria, membrane damage, dissolution of internal cristae, etc. (Yu et al., 2011; Wang, Zhou & Mao, 2020; Das, 2021; Li et al., 2022). Similarly, as an important site for photosynthesis, chloroplasts are destroyed under drought, which shows that the photosynthesis of the plant itself has been inhibited, resulting in a decrease in photosynthetic capacity. The osmiophiles are droplets formed by aggregation of some lipid substances, which are used as storage of lipid substances in plant chloroplasts (Huan et al., 2014). These osmiophiles are supposed to act as an electron carrier protecting the vesicle from free radical damage caused by adversity stress (Grigorova et al., 2012). In addition, it is also considered to be the product of polymerization after thylakoid degrades the membrane lipid. In this study, the number of osmiophiles increased under moderate drought, which also showed that the coniferous chloroplasts of P. sylvestris var. mongolica seedlings had been damaged under moderate drought. The results of mitochondrial research were consistent with the previous studies on Fraxinus mandshurica, Ormosia hosiei, and Cyclocarya paliurus seedlings (Wei, Wang & Zhang, 2010; Liu & Wei, 2019; Li et al., 2022). As the main site of the tricarboxylic acid cycle and oxidative phosphorylation, mitochondria were destroyed under drought stress, showing that needle cells struggle to maintain normal physiological functions, resulting in changes in needle structure.

Conclusions

The plant height, ground diameter, biomass, and photosynthetic pigment indexes of P. sylvestris var. mongolica seedlings decreased with increasing degrees of drought stress. Drought stress inhibited total root length, root surface area, root volume, total root length per unit of soil volume, bifurcation number, and crossing number of P. sylvestris var. mongolica seedlings. Furthermore, the deepening of the drought caused the stomatal density to increase first, followed by a decrease. The drought stress also led to changes in the structure of roots, stems, needles, needle cell structure, and organelle structure of P. sylvestris var. mongolica seedlings, thus, destroying the normal metabolic pathway and physiological function of cells. Morphological changes in organelles also provide cytological evidence for the study of drought resistance in P. sylvestris var. mongolica. In summary, P. sylvestris var. mongolica is affected by the increasing drought pressure caused by current climate change, making its risk of recession under drought higher. Therefore, special attention should be paid to the effect of environment on Mongolian pine when afforestation is carried out in arid and semi-arid areas.

Supplemental Information

Data S1 Raw data

Click here for additional data file.

Additional Information and Declarations

Competing Interests

Author Contributions

Data Availability

The authors declare there are no competing interests.

Fanjun Meng conceived and designed the experiments, performed the experiments, analyzed the data, prepared figures and/or tables, authored or reviewed drafts of the article, and approved the final draft.

Tianze Zhang conceived and designed the experiments, performed the experiments, analyzed the data, prepared figures and/or tables, authored or reviewed drafts of the article, and approved the final draft.

Dachuan Yin performed the experiments, analyzed the data, prepared figures and/or tables, authored or reviewed drafts of the article, and approved the final draft.

The following information was supplied regarding data availability:

The raw measurements are available in the Supplementary File.

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
