# Peer review of "The effects of soil drought stress on growth characteristics, root system, and tissue anatomy of Pinus sylvestris var. mongolica"

_PeerJ, doi:10.7717/peerj.14578_

## Round 0.1 · original submission · Major Revisions

Dear Authors,

Three independent experts assessed your work. Everyone agreed that the work could be published in PeerJ, but it should be significantly improved beforehand. Please read the comments of the reviewers carefully and respond to all of them.

With best regards,

Reviewer 1 ·

Basic reporting

I really enjoyed read the manuscript titled “Effects on growth characteristics, root system, and tissue anatomy of Pinus sylvestris var. mongolica under soil drought stress”. It evidenced the effect of drought effect on seedlings in lab and controlled conditions on large tree characteristics from roots to needles. The authors found that damage was directly correlated to drought stress intensity.
The paper was easy reading but (as not native English speaker) I have some doubts about the English language. I found sometimes redundant structures in sentence (for example lines 35-36), so I would recommend to certify that the article is reviewed by a fluent English speaker.
In the introduction section, I missed more references of drought experiments for forest trees (line 79-81), especially for Scots pine (I'm sure references can be easily found). I also missed some cites at the end of the sentences, for example in lines 42-44,63-66, 75-77, 91-92…
In my opinion, there are a lot of figures, so I would recommend to move some of them to supplementary material for an easy reading (for example Fig.1 or Fig 3. which is almost redundant with Fig.4, and move Fig 7 too). Continuing with the figures, I would recommend include in Fig. 9 several white arrows to make clear what you want show. It is strange for me, to include a picture in Discussion section. I would recommend to add that figure (Fig. 10) as graphical abstract, but please, modify the word “change” for “increase” or “decrease” for a clear statement about drought effect.
Raw data supplied is fine, but I found strange that authors mentioned in the text to measure 10 pots by treatment and 8 seedlings per pot (lines 120-122), i.e., 80 sample seedlings by treatment (?) and in the data can only be found 8-6 (or sometimes lower) measurements (rows in the excel database). The number of needles analysed for stomatal density and chlorophyll content must be indicated also in the text.

Experimental design

The original research matches with the scope of the journal, and the research question is well defined and relevant. Methods are also described with sufficient detail. However, as mentioned, previously I am not sure about the total number of seedlings analysed per treatment. In addition, if more than one seedling is included by pot, I wondering if pot factor should be included in the statistical analysis in which seedlings were nested in. If I had to do this experimental design, I would use a pot by seedling with a large number of seedlings by treatment (maybe 30) and repeat treatment at least 3 times, so a block effect could be considered for example in a mixed model. In this way, results variance will decrease and even more clear findings would arise. Anyway, the number of seedlings analysed here could be enough to find some results.

Validity of the findings

Conclusions are well stated, linked to original research question & limited to supporting results. Replication was enough but could be improved to obtain clearer results. Differences from Fig.2 (lines 185-187) are no possible to validate if SE or interval coefficient are not shown. Letters showing significant differences and error bars from figures sometimes seems to describe different findings (this is especially true in Fig. 6).

Additional comments

• Please move lines 255-260 to introduction
• I missed to compare the results with other S.pine drought experiments in discussion section (and include them in introduction too)
• Reference list is not alphabetically shorted

Reviewer 2 ·

Basic reporting

The introduction and discussion need some work.
Showing in the introduction the literature on sunflower, for example, is a major understatement (l. 80-81). Here should be literature showing the response of forest trees to drought stress. A similar comment applies to the discussion. The introduction should also include newer publications than the 2013 IPCC data (l.38).I propose to change the title as well, the current one suggests that the research was conducted on large, adult trees.
In my opinion manuskrypt is use clear, unambiguous, technically correct English language, but I don't feel qualified to judge about the English language and style

Experimental design

no comment

Validity of the findings

In the manuscript (Results) is no need to provide values. Is imprtant, if the trend is rising or falling. Is ok e.g. lines 223-225, 241-247. The reader will not remember that e.g. l. 214-216

Additional comments

Below I present my comments on individual parts of the manuscript:

l. 38: Please refer to later reports IPCC than 2013
l. 36 - 37 :„Climate change has caused various environmental concerns apparently, such as drought caused by climate change, which affects the worldís biological distribution (Sherwood et al.,37 2013; Trenberth et al., 2014)”- the use of the term "climate change" twice in one sentence makes it unclear to the reader
l. 43-44: In China, the arid and semi-arid land can reach half or more of the land) - in what time perspective can this happen? There is no literature data to support this statement

l. 64-66: „In the process of growth, plants start their drought response mechanism to cope with water shortage, through the change of their morphological structure, the synthesis of osmotic adjustment substances, photosynthetic pigments, hormones, and the expression of drought resistance genes” - this sentence is more of a textbook for students, it is very general, it does not give any specific information,
l.68: „the root structure determines the water absorption and transport efficiency of plants, which
can help them to alleviate the damage caused by drought stress”- mycorrhiza cannot be forgotten either

l. 80-81 : „Some studies found that mild drought stress increased the root length of sunflowers,
whereas the growth of root length decreased with a further increase in drought stress
(Manivannan et al., 2007)” - please show forest tree research, not sunflower

Material and methods:

Why each pot (of seedlings ) was weighed every 128 day to supplement the lost water?
There is no information on how long the experiment lasted

Results
General comment on part of Results: The manuscript also fails to show the most common differences, which are statistically insignificant. There is no need to provide values that give the exact way in which drought changes are taking place. The trend is whether it is rising or falling.
Is ok e.g. lines 223-225, 241-247. The reader will not remember that e.g. l. 214-216 („There was a decrease of 18.18% and 13.64% under MD treatment and 36.36% and 27.27% under HD treatment, respectively, in the dry weight of primary root and lateral root (Fig. 4A). Furthermore, the dry weight of the stem decreased by 14.29% and 21.43% compared with CK under MD treatment and HD treatment (Fig. 4B), and the leaf decreased by 8.33% and 31.25% compared with CK under MD treatment and HD treatment, respectively (Fig. 4C)”.

l.183-185: twice repeating the information on what is presented in Fig. 1.
l. 234: the term "decreasing trend, though the overall change was not obvious" is unclear.
l. 257 : In this part of the work, we present our results and do not refer to the literaturę.
l. 255-260, 263-265 - this parts are not the results
l.318: „Poorter H”- letter H unnecessary

Discussion
Please refer to the works showing drought stress in trees and not in fescue or Halogeton arachnoideus in the discussion. There is really a lot of work in this area.
l..416: The authors write “A large number of studies” but they provide only one publication.

Reviewer 3 ·

Basic reporting

The article is written in clear and correct English.
The article includes a sufficient introduction but I recommend underlining the aim of the study and a few general research questions or presenting clear hypotheses(s).
The relevant literature is appropriately referenced
The structure of the manuscript is appropriately and fits scientific standards.
Figures are relevant to the content and sufficiently described.
Raw data were not included.

Experimental design

The topic of the study fits the Aims and Scope of the journal.
Research gaps should be stated in a more precise way.
I missed some research gaps to be mentioned in the introduction.
The research was conducted in a proper and scientific sound way.
Methods were described in a sufficient way.

Validity of the findings

The scientific side of the experiment and the possibility to replicate it is fulfilled.
The raw data are not included.
The weak side of the text is the section "Conclusions". This is rather a summary of the results, not conclusions. Conclusions must emerge from the results of the research eg. what are the conclusions for forest management?

Additional comments

The manuscript's topic touches on a significant issue: the influence of drought on tree species growth. It is very important to analyze the question in real field conditions, however, also pot cultivation for the seedlings, kept in artificial conditions, can be used as well. The authors applied the second method to reveal the reaction of P. sylvestris var. mongolica seedlings to drought.
I found the manuscript interesting but it needs some improvements.

I suggest changing the title to "The effects of soil drought stress on growth characteristics, root system, and tissue anatomy of Pinus sylvestris var. mongolica" OR
"Growth characteristics, root system, and
tissue anatomy of Pinus sylvestris var. mongolica under soil drought stress"

Abstract: the sentence starting with "The indoor culture continuous pot....." should be moved earlier, after the sentence ending "....pot cultivation"

Keywords shouldn't repeat words included in the title. Please change them.
Line 36: "caused by climate change" is a repetition of the information given earlier in the same sentence
Line 39: delete "in the world"
Line 42-44: the sentences are not connected to the previous and next paragraphs. I suggest deleting them.
Line 47: what does "early timber" mean? It needs explanation
In all text please change "leaves" to "needles". P. sylvestris has needles, not leaves. In many places of the text, the Authors used "leaves" instead of "needles"
Line 179: if analysis of variance is going to be used. it is necessary to check the normality of the variable's distribution. Has it been done?
Line 183-184: delete "treated in each treatment"
Line 189: use "state" instead of "treatment"
Line 191: delete "at the beginning"
Line 191: ....increased only by....
Line 198: use "water deficit" instead of "drought percentage"
Figure 3,4: please change from "leaves" to "needles"
Line 209-211: in Fig 3a something different is visible. Please check the text with the figure.
Line 255-257: the sentence fits better in the introduction, not to results
Line 342: I assumed that it should be "In this process, chlorophyll a is mainly responsible for converting light energy into chemical energy."
Line 344-346: some references are needed
Line 416-419: "A large number of studies...." but only one reference is provided ?? (Yu et al,. 2011)
Section Conclusions should be rewritten completely. See my previous comments.
References should be in alphabetical order.
Figure 10: I suggest giving more accurate information, It means instead of "change" put "decrease" or "increase" in the graph.
Table 1: The table must be self-explained. So the acronym "LenPerVol" should be explained.

---

## Round 0.2 · Minor Revisions

Dear Authors,

Your work was re-assessed by 3 experts. One of them still has comments about your work. I kindly ask you to read these remarks and make appropriate changes to the manuscript.

With best regards,

Reviewer 1 ·

Basic reporting

It is noted that the authors have made a great effort answering the reviewers and the article has been considerably improved and meets all the basic requirements.

Experimental design

It is still no clear for me the total number of plants analyzed by treatment.

Validity of the findings

Results and Conclusions are well stated and interesting.

Additional comments

I think it could be accepted in the journal for publication since the authors did a big effort answering the reviewers' comments so the paper improved significantly.

Reviewer 2 ·

Basic reporting

I am happy to read the manuscript again. It is now much easier to read and clearer. There is a reference in the text about how drought affects other forest tree species (which I missed in the first version).

It is strange for me, to include a picture in Discussion section (Fig.10).

In my opinion manuskrypt is use clear, unambiguous, technically correct English language, but I don't feel qualified to judge about the English language and style.

The weak side of the text is the section "Conclusions" still. This is rather a summary of the results, not conclusions. Conclusions must emerge from the results of the research eg. what are the conclusions for forest management? I still don't see that.

Experimental design

The original research matches with the scope of the journal, and the research question is well defined and relevant. Methods are also described with sufficient detail

Validity of the findings

The scientific side of the experiment and the possibility to replicate it is fulfilled.

Additional comments

none is Correspondence: e-mail:
l.78-91- is crossed out
l.376- Nina T. – „T” isn't needed
l.400- space before the comma in „Japanese Cedar”
l.399-400- please give the Latin name "Japanese Cedar”
letter in name eg. l.381, 389, 401,
in references there are minor errors, e.g. sometimes there is a parenthesis next to the year and sometimes there is not, space before the period at the end

Reviewer 3 ·

Basic reporting

Nothing to add

Experimental design

Nothing to add

Validity of the findings

Nothing to add

Additional comments

I found that the Authors integrated all my suggestions and comments into the revised version of the manuscript.
I have nothing to add.
Nice job.

---

## Round 0.3 · accepted · Accept

Dear Authors - congratulations! All reviewers agreed that your work can be published in PeerJ in its current version.

With best regards,

Reviewer 2 ·

Basic reporting

I am happy to read the manuscript again. It is now much easier to read and clearer. I found that the Authors integrated all my suggestions and comments into the revised version of the manuscript.

Experimental design

no comment

Validity of the findings

The scientific side of the experiment and the possibility to replicate it is fulfilled.

Additional comments

The manuskrypt could be accepted in the journal for publication, the authors did a big effort taking into account the reviewers' comments in the amendments